# BENCHMARKING LARGE LANGUAGE MODEL BENCH-MARKS: POPULAR BENCHMARKS VS. HUMAN PERCEPTION

## ABSTRACT

Benchmarks play a critical role as a measure of large language model (LLM) capabilities. However, whether LLM performance on benchmarks is similar to their real-world performance, especially human perception of their outputs, remains questionable. This study specifically focuses on whether **LLM performance on benchmarks is similar to human perception**. The study investigates this gap by quantifying the similarity between LLM rankings derived from benchmarks and LLM rankings generated from human votes on the prominent LMArena platform. It systematically compares benchmark rankings against rankings in corresponding task-specific categories in LMArena for over 100 top-tier LLMs. The findings reveal that LLM performance on several popular benchmarks has low similarity with human perception, even though these benchmarks are more recent or challenging. The results highlight limitations in current benchmarking practices and underscore the need for evaluation frameworks that more accurately reflect the human perception and real-world performance of LLMs.

## 1 INTRODUCTION

Benchmarking has become a cornerstone for evaluating large language models (LLMs). When new LLMs are released, their capabilities are typically assessed and compared using standardized benchmarks, such as (Hendrycks et al., 2021b;a), LiveCodeBench (Jain et al., 2025), and Humanity's Last Exam (Phan et al., 2025). These benchmarks offer a convenient framework for quantifying LLM performance. In recent months, both open-source models and closed-source models have reported state-of-the-art (SOTA) or near-SOTA results on major benchmarks. Examples of these open-source models include the Qwen 3 series (Yang et al., 2025), DeepSeek V3.1 (DeepSeek, 2025), and Kimi-K2 (Team et al., 2025a). Examples of these closed-source models include GPT-5 (OpenAI, 2025), Gemini 2.5 Pro (Comanici et al., 2025), and Claude Opus 4.1 (Anthropic, 2025).

However, as Sam Altman has observed, users increasingly care *less* about benchmark scores and *more* about who is using an LLM and what value they derive from it (Theo Von, 2025). This suggests that despite the rapid progress reflected in SOTA benchmark results, the gap between human perception of LLM outputs and benchmark performance is widening, and trust in benchmark results is declining (Pandey, 2025). These issues are often attributable to cases where AI systems that appear impressive in benchmark evaluations perform poorly in real-world applications (Petrosino, 2025). A likely reason is that models, researchers, and vendors often optimize for the benchmark rather than the real task (Liubimov, 2025). The above observations lead to our central research question:

**How does LLM performance on benchmarks differ from human perception?**

To explore this, we searched for a benchmark that is fully grounded in real human perception. We identified *LMArena* (Zheng et al., 2023), an open platform where users interact with and compare leading LLMs. By allowing side-by-side comparisons and collecting votes for the better response, the platform enables the community to help rank over 250 LLMs. Additionally, LMArena assesses LLMs' strengths and weaknesses in a more granular way by grouping user-submitted prompts into

task-specific categories, such as math prompts and coding prompts. LMArena provides a ranking of LLMs for each category of prompts.

Benchmarks can also be classified according to the skill they measure, like mathematical reasoning or coding ability. Given this categorization, our approach is centered on a key assumption: If a benchmark's ranking of LLMs differs from the ranking of LLMs on the corresponding category in LMArena, then LLM performance on that benchmark must differ from human perception. Guided by this assumption, we compare each benchmark's ranking against rankings in corresponding task-specific categories in LMArena.

We draw the following conclusions from the above comparison:

- We found that some popular benchmarks have rankings not similar to LMArena's, like *Humanity's Last Exam*, *FACTS Grounding*, and *IFEval*. LLM performance on these benchmarks may differ from human perception.

- LLM performance on neither more difficult nor easier benchmarks is necessarily more similar to human perception.

- LLM performance on newer benchmarks is not necessarily more similar to human perception, except for benchmarks that evaluate instruction following or the creative writing ability of LLMs. These benchmarks are being improved to better resemble human perception of LLM outputs.

- Compared with other abilities (e.g., coding ability and instruction following ability), the math ability of LLMs can be relatively better evaluated by current benchmarks.

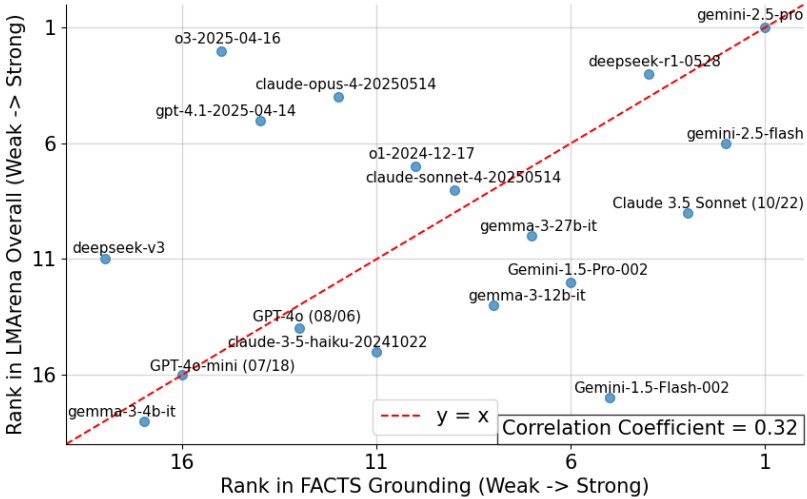

Figure 1: Comparison of large language model (LLM) ranking in FACTS Grounding and the overall ranking in LMArena. Each point $(x_i, y_i)$ represents an LLM ranked $x_i$-th in $A'$ and $y_i$-th in $B'$, where $A'$ and $B'$ are adjusted rankings based only on LLMs that appear in both original rankings $A$ and $B$. In this case, $A$ and $B$ represent the ranking in FACTS Grounding and the overall ranking in LMArena, respectively. The Spearman rank-order correlation coefficient of these two rankings is merely 0.32. We can conclude that LLM performance on FACTS Grounding may differ from human perception.

These findings expose both the strengths and limitations of popular benchmarks and underscore the urgent need for more representative benchmarks—especially for open-ended, creative, and user-centric tasks.

## 2  RELATED WORK

Among the benchmarks we investigated, some use LLMs as judges. Most recent work on benchmarking focused on the reliability of LLMs as judges. Zheng et al. (2023) found that powerful LLM judges like GPT-4 can achieve over 80% agreement with human evaluations. However, Wang et al. (2024a) revealed systematic biases when using LLMs as judges, indicating flaws in this evaluation paradigm. Furthermore, Koo et al. (2024) thoroughly investigated the correlation between human preferences and machine preferences, calculating an average Rank-Biased Overlap (RBO) score of just 44%, indicating significant deviation between machine and human preferences. Li et al. (2025a) noted that current LLM evaluators still have certain limitations. However, they also found that when LLMs carefully consider various criteria before giving overall scores, they can achieve higher correlation with human assessments. Nonetheless, none of the aforementioned work has focused on whether benchmark results that do not use LLMs as judges are similar to human perception.

Before LLMs became prevalent, researchers studied the reliability of benchmarks that do not use LLMs as judges, but these benchmarks were not used to evaluate LLMs either. Kiela et al. (2021) pointed out that while contemporary models perform well on benchmarks, they still underperform on simple challenge cases and real-world scenarios, reflecting a disconnect between benchmark performance and practical needs. Dehghani et al. (2021) revealed significant variations in model rankings across different benchmarks. For example, the average Kendall's rank correlation between subtasks and composite scores in the SuperGLUE benchmark was only 0.648, suggesting that even different tests within the same benchmark may produce inconsistent evaluations, and the correlation between subtasks and average results is imperfect. However, most of the benchmarks focus on performance averaged over many tasks, and the question of how to reliably evaluate and tune models trained for individual tasks in this regime has not been addressed (Shimabucoro et al., 2024). Meanwhile, Liang et al. (2023) studied the correlation between model rankings on accuracy, robustness, and fairness. Yet, they did not explore the relationship between accuracy and human perception of model outputs.

## 3  METHOD

We quantify the similarity between LLM rankings derived from benchmarks and rankings collected through LMArena by calculating the correlation between them. This section addresses the following questions:

1. Which LLMs do we investigate?
2. Which rankings from benchmarks do we investigate?
3. Which rankings from the LMArena platform do we investigate?
4. How do we calculate the correlation between two rankings?

### 3.1  INVESTIGATED SET OF LLMS

Since powerful LLMs with larger parameter sizes often receive more attention and are most frequently employed for human-facing services, we focus our investigation specifically on the set of LLMs that have scores not lower than 1300 on the overall LMArena leaderboard. This set of LLMs has a size of 114, and in other words, we investigate the LLMs ranked top 114 on the overall LMArena leaderboard.

### 3.2  INVESTIGATED LLM RANKINGS FROM BENCHMARKS

We choose 24 public benchmarks on the basis of the wide adoption by top-tier LLMs today, and collect the rankings of the above LLMs from benchmark leaderboards.

We organize our benchmarks into four categories: Question Answering (QA), Mathematics, Code, and Alignment. This classification reflects the core skills emphasized in evaluations for state-of-the-art models such as GPT-5 (OpenAI, 2025), Gemini 2.5 Pro (Comanici et al., 2025), Qwen 3 (Yang et al., 2025), and DeepSeek-R1 (DeepSeek-AI et al., 2025), as well as the organizational structure of widely recognized leaderboards like the Open LLM Leaderboard (Aidar Myrzakhan, 2024). We also draw on insights from a recent comprehensive survey on LLM evaluation (Cao et al., 2025).

Table 1: Investigated benchmarks and their categorization.

| Question Answering | Mathematics | Code | Alignment |
|---|---|---|---|
| GPQA | MGSM | HumanEval | IFEval |
| MMLU-Pro | MATH-500 | LiveCodeBench | Arena-Hard |
| SimpleQA | FrontierMath | SWE-Bench Verified | WritingBench |
| FACTS Grounding | AIME | Aider Polyglot | Creative Writing v3 |
| Humanity's Last Exam | HMMT February 2025 | Terminal-Bench | IFBench |
| SuperGPQA | | SciCode | |
| ARC-AGI-2 | | IOI | |

1. **Question Answering**: Evaluates question answering ability, including factual knowledge recall and logical reasoning. This category includes 7 benchmarks: *GPQA* (Rein et al., 2024), *MMLU-Pro* (Wang et al., 2024b), *SimpleQA* (Wei et al., 2024), *FACTS Grounding* (Jacovi et al., 2025). *Humanity's Last Exam* (Phan et al., 2025), *SuperGPQA* (Team et al., 2025b), *ARC-AGI-2* (Chollet et al., 2025).

2. **Mathematics**: Evaluates mathematical problem-solving ability. This category includes 5 benchmarks: *MGSM* (Shi et al., 2023), *MATH-500* (Lightman et al., 2024), *FrontierMath (Tier 1-3)* (Glazer et al., 2025), *AIME* (AoPS, 2025), *HMMT February 2025* (Harvard–MIT Mathematics Tournament (HMMT), 2025).

3. **Code**: Evaluates code generation and understanding ability. This category includes 7 benchmarks: *HumanEval* (Chen et al., 2021) *LiveCodeBench* (Jain et al., 2025), *SWE-bench Verified* (Jimenez et al., 2024; Chowdhury et al., 2024), *Aider Polyglot* (Aider, 2024), *Terminal-Bench* (Team, 2025), *SciCode* (Tian et al., 2024), *IOI* (International Olympiad in Informatics, 2025).

4. **Alignment**: Evaluates instruction following and creative writing ability. This category includes 5 benchmarks: *IFEval* (Zhou et al., 2023), *IFBench* (Pyatkin et al., 2025) *ArenaHard* (Li et al., 2025b), *WritingBench* (Wu et al., 2025), *Creative Writing v3* (Paech, 2025).

Table 1 summarizes the benchmarks within each of our four categories. Descriptions of these benchmarks and the sources of the ranking data on these benchmarks can be found in the appendix, in Table 7. Table 9 shows the benchmarks we investigated but not adopted, and the reasons can also be seen in the appendix.

## 3.3 INVESTIGATED LLM RANKINGS FROM LMARENA

Correspondingly, besides the overall ranking in LMArena, LMArena groups human prompts into various categories, and for each category, it generates an LLM ranking. Each ranking reflects human perception of how well LLMs respond to prompts within this category. To compare the benchmark rankings more rigorously, in addition to the **overall ranking in LMArena**, we investigate three other categories in LMArena, each with its own ranking: **Coding**, **Math**, and **Instruction Following**. More details can be found in the LMArena Blog (https://news.lmarena.ai/chatbot-arena-categories).

We collect the scores and rankings of LLMs in the set of LLMs defined in Section 3.1 on LMArena.

## 3.4 CORRELATION COMPUTATION

Given that our rankings are derived from benchmark scores with heterogeneous distributions, we compute the Spearman rank-order correlation coefficient (Virtanen et al., 2025) between rankings. The Spearman correlation is a non-parametric statistic that assesses the strength and direction of association between two variables without requiring assumptions of linearity or normality.[1]

A ranking is an ordered list of LLMs sorted by score or accuracy. Suppose we have two rankings:

$$A = [a_1, a_2, a_3, \ldots, a_n], \quad B = [b_1, b_2, b_3, \ldots, b_m],$$

---

[1]We do not use the Pearson correlation coefficient, which is a parametric measure that presumes approximate normality of the variables involved.

where each element represents an LLM.

We then remove all LLMs in $A$ that do not appear in $B$ to obtain $A'$. Similarly, we remove all LLMs in $B$ that do not appear in $A$ to obtain $B'$. The resulting lists $A'$ and $B'$ have the same length and contain the same set of LLMs, denoted as $A \cap B$. The number of LLMs in $A \cap B$ is defined as $N \triangleq |A \cap B|$, representing the number of data points involved in computing the correlation coefficient.

In a Cartesian coordinate system (refer to Figure 1), each LLM in $A \cap B$ is plotted as a point $(x_k, y_k)$, where $x_k$ is the rank of that LLM in $A'$ and $y_k$ is the rank in $B'$. Using these $N$ pairs $(x_1, y_1), (x_2, y_2), \ldots$, we compute the Spearman correlation coefficient $\rho$ as

$$\rho = 1 - \frac{6\sum_{k=1}^{N}(x_k - y_k)^2}{N(N^2 - 1)}.$$

In practice, we use the library function `scipy.stats.spearmanr` from the Python library `scipy` to calculate $\rho$. For significance (p-value), when $N < 30$, we apply a permutation test with `scipy.stats.permutation_test` and `permutation_type='pairings'`, which permutes one ranking under the null hypothesis of independence and recalculates the correlation. When $N \geq 30$, the p-value returned by `scipy.stats.spearmanr` is used as a sufficiently accurate approximation.

In Section 3.3, we collect 4 distinct rankings from LMArena and its categories. Let $\{A_1, A_2, A_3, A_4\}$ denote these rankings, and $\{B_1, B_2, \ldots, B_{24}\}$ denote the rankings from the 24 benchmarks. For each pair $(A_i, B_j)$ in the Cartesian product of the above two sets, we compute $\rho(A_i, B_j)$, as shown in the schematic table (Table 2): rows represent $B_1$–$B_{24}$, columns represent $A_1$–$A_4$, and we compute $\rho(A_i, B_j)$ in each cell. All correlation coefficients are reported along with their p-values.

Table 2: A schematic table showing Spearman correlation coefficients $\rho(A_i, B_j)$ calculated between different task-specific rankings in LMArena $\{A_1, A_2, A_3, A_4\}$ and twenty-four benchmark rankings $\{B_1, B_2, \ldots, B_{24}\}$. For the computed value of each $\rho(A_i, B_j)$, please refer to Table 5 in the appendix.

|          | $A_1$             | $A_2$             | $A_3$             | $A_4$             |
|----------|-------------------|-------------------|-------------------|-------------------|
| $B_1$    | $\rho(A_1, B_1)$  | $\rho(A_2, B_1)$  | $\rho(A_3, B_1)$  | $\rho(A_4, B_1)$  |
| $B_2$    | $\rho(A_1, B_2)$  | $\rho(A_2, B_2)$  | $\rho(A_3, B_2)$  | $\rho(A_4, B_2)$  |
| $\vdots$ | $\vdots$          | $\vdots$          | $\vdots$          | $\vdots$          |
| $B_{24}$ | $\rho(A_1, B_{24})$ | $\rho(A_2, B_{24})$ | $\rho(A_3, B_{24})$ | $\rho(A_4, B_{24})$ |

Benchmarks only focus on the LLMs' performances in the specific skill they measure, like mathematical or coding ability. These skills, measured by benchmarks, relate to distinct task-specific rankings in LMArena. So we focus on the correlation coefficients between benchmark rankings and rankings in corresponding task-specific categories in LMArena. In detail, we pay more attention to the similarity between LLM rankings on benchmarks in the QA category with the overall ranking in LMArena, on benchmarks in the Mathematics category with the LMArena Math category, on benchmarks in the Code category with the LMArena Coding category, and on benchmarks in the Alignment category with the LMArena Instruction Following category.

## 4 RESULTS

This section presents the results of our research. Section 4.1 analyzes the correlation between benchmark rankings and rankings in corresponding task-specific categories in LMArena. We observe that some popular benchmarks have rankings not similar to LMArena's. To further explore this, we investigate how the correlation coefficient relates to other factors—specifically, difficulty levels of benchmarks in Section 4.2 and release dates of benchmarks in Section 4.3. Finally, for each task-specific category in LMArena, Section 4.4 computes the average correlation coefficients across all associated benchmarks, concluding the section.

## 4.1 CORRELATION ANALYSIS OF INDIVIDUAL BENCHMARKS

The four tables Tables 3a to 3d present the correlation between each benchmark ranking and the ranking on the corresponding task-specific category in LMArena.

Table 3a shows that LLM ranking on FACTS Grounding shows no significant correlation with the overall ranking in LMArena, marked as "N.S.". It also shows the lowest correlation ($\rho = 0.32$). Rankings on Humanity's Last Exam also show no significant correlation, ranking second lowest in Table 3a. Table 3b shows that rankings on all mathematics benchmarks have a significant correlation with LMArena's Math ranking of LLMs. Table 3c shows that ranking on IOI shows the lowest correlation ($\rho = 0.62$) with LMArena's Code ranking of LLMs. Table 3d shows that ranking on IFEval shows a weak correlation ($\rho = 0.45$) with LMArena's Instruction Following ranking of LLMs. Ranking on IFBench also shows a low correlation ($\rho = 0.61$, ranking second lowest in Table 3d). We noticed that both IFEval and IFBench use predefined rules for evaluation. Conversely, Creative Writing v3, Arena-Hard, and WritingBench use LLMs as judges. This suggests that Alignment benchmarks that use LLMs as judges may be better at resembling human perception.

**Conclusion:** Some popular benchmarks have rankings not similar to LMArena's, like Humanity's Last Exam, FACTS Grounding, and IFEval. LLM performance on these benchmarks may differ from human perception.

Table 3: The Spearman correlation coefficient ($\rho$) between each benchmark ranking and the ranking on the corresponding task-specific category in LMArena is presented in the table below. The table is organized into four sub-tables according to the type of benchmarks and their corresponding categories. Specifically, sub-table (a) shows the correlations for QA benchmarks against the overall ranking in LMArena; sub-table (b) for Mathematics benchmarks against the Math category in LMArena; sub-table (c) for Code benchmarks against the Coding category in LMArena; and sub-table (d) for Alignment benchmarks against the Instruction Following category in LMArena. $N$ represents the number of LLMs that are common to both rankings being compared when calculating the Spearman correlation. For statistical significance, we use the p-value with thresholds: ***$p < 0.001$, **$p < 0.01$, *$p < 0.05$; N.S. = Not significant ($p \geq 0.05$).

(a) QA vs LMArena's overall ranking

| Benchmark | $\rho$ | $N$ | Significance |
|---|---|---|---|
| SuperGPQA | 0.92 | 27 | *** |
| MMLU-Pro | 0.86 | 42 | *** |
| SimpleQA | 0.79 | 23 | *** |
| GPQA | 0.79 | 28 | *** |
| ARC-AGI-2 | 0.71 | 18 | ** |
| Humanity's Last Exam | 0.51 | 12 | N.S. |
| FACTS Grounding | 0.32 | 18 | N.S. |

(b) Mathematics vs LMArena's Math ranking

| Benchmark | $\rho$ | $N$ | Significance |
|---|---|---|---|
| MATH-500 | 0.89 | 23 | *** |
| FrontierMath | 0.87 | 29 | *** |
| HMMT February 2025 | 0.85 | 18 | *** |
| AIME | 0.78 | 29 | *** |
| MGSM | 0.71 | 27 | *** |

(c) Code vs LMArena's Code ranking

| Benchmark | $\rho$ | $N$ | Significance |
|---|---|---|---|
| Aider polyglot | 0.87 | 25 | *** |
| SWE-bench Verified | 0.8 | 13 | ** |
| Terminal-Bench | 0.8 | 49 | *** |
| HumanEval | 0.78 | 59 | *** |
| LiveCodeBench | 0.77 | 14 | ** |
| SciCode | 0.68 | 69 | *** |
| IOI | 0.64 | 10 | * |

(d) Alignment vs LMArena's Instruction Following ranking

| Benchmark | $\rho$ | $N$ | Significance |
|---|---|---|---|
| Creative Writing v3 | 0.84 | 26 | *** |
| Arena-Hard | 0.73 | 18 | *** |
| WritingBench | 0.72 | 25 | *** |
| IFBench | 0.61 | 47 | *** |
| IFEval | 0.45 | 20 | * |

The conclusion above raises a critical question: Why do certain benchmark rankings—such as Humanity's Last Exam—show such low correlation with human perception? A hypothesis is that these benchmarks often test very difficult skills. For example, Humanity's Last Exam poses extremely complex problems aimed at AGI-level reasoning. Given the observed low correlation for notoriously difficult benchmarks, a natural question arises: What is the relationship between benchmarks' difficulty and correlation coefficient with the corresponding LMArena category? The following section explores the problem.

## 4.2 RELATIONSHIP BETWEEN BENCHMARK DIFFICULTY AND CORRELATION WITH HUMAN PERCEPTION

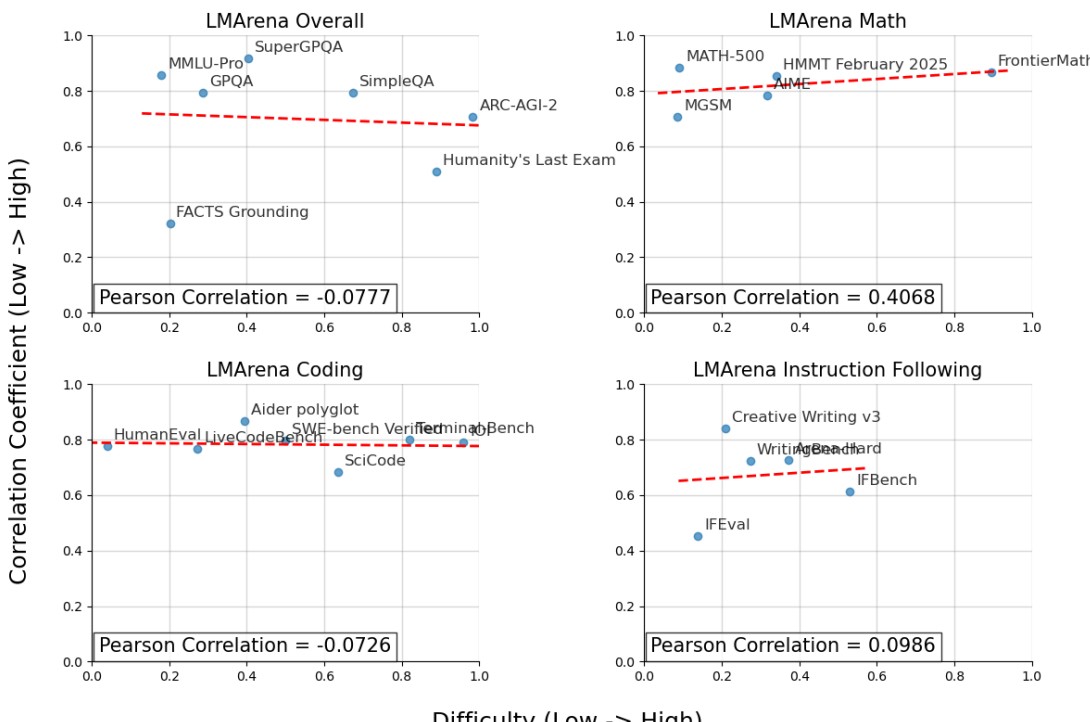

Figure 2: Relationship between benchmark difficulty and the Spearman correlation coefficient ($\rho$) with the corresponding LMArena category. The horizontal axis represents the average error rate of LLMs in a fixed set on the benchmark, where a higher value indicates greater difficulty. Each point $(x_i, y_i)$ corresponds to a benchmark with an average error rate $x_i$, and $y_i$ is the correlation coefficient between the LLM ranking on that benchmark and the ranking of the LMArena category shown above the subfigure. The red line is the regression line, and the indicated number is the Pearson correlation coefficient between $x$ and $y$.

As discussed in the Evaluation sections of *Humanity's Last Exam* (Phan et al., 2025) and *Auto-Bencher* (Li et al., 2025c), the accuracy achieved by top-tier LLMs on a benchmark can serve as an indicator of the benchmark's difficulty. To assess this difficulty more rigorously, we compute the average error rate, defined as one minus the average accuracy of a fixed set of LLMs. In our study, this set comprises models whose overall scores on the LMArena leaderboard fall between 1386 and 1418. We thus define benchmark difficulty as the average error rate. A higher difficulty value indicates a more challenging benchmark, which is likely to remain useful for evaluating future LLMs. Conversely, a lower difficulty suggests that the benchmark is nearing saturation and may have limited utility in discriminating among top-tier LLMs.

Figure 2 shows that benchmark difficulty does not strongly influence the Spearman rank-order correlation coefficient. This observation is consistent across all four categories of benchmarks, challenging a common assumption that more challenging benchmarks better reflect real-world performance. Our results do not support this view. This highlights the need for careful benchmark design that considers human preferences directly.

**Conclusion:** LLM performance on neither more difficult nor easier benchmarks is necessarily more similar to human perception.

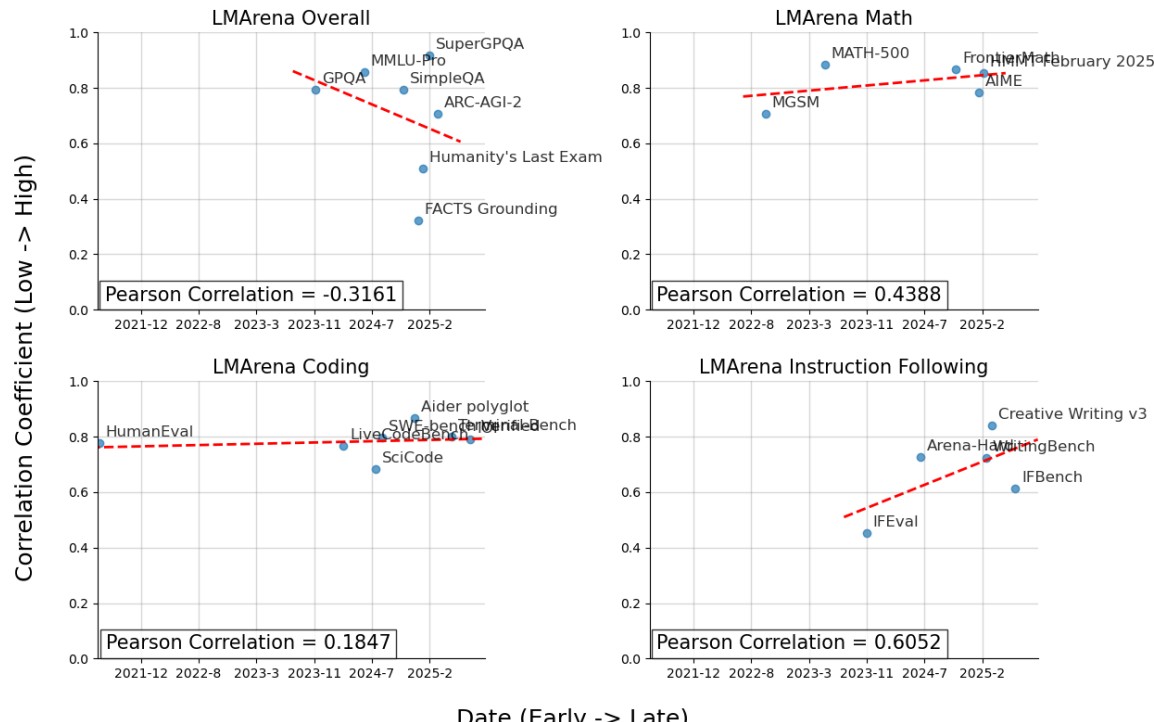

Figure 3: Relationship between benchmark publication date and the Spearman correlation coefficient ($\rho$) with the corresponding LMArena category. Each point $(x_i, y_i)$ represents the benchmark that was released at date $x_i$, and $y_i$ is the correlation coefficient between its ranking and the ranking of the LMArena category shown above the subfigure. The red line is the regression line, and the indicated number is the Pearson correlation coefficient between $x$ and $y$.

### 4.3 RELATIONSHIP BETWEEN BENCHMARK RECENCY AND CORRELATION WITH HUMAN PERCEPTION

Figure 3 examines the relationship between the correlation coefficients of all benchmarks with the rankings on corresponding LMArena categories, and the initial release dates of these benchmarks. We create four scatter plots, with the horizontal axis representing the initial release date of benchmarks and the vertical axis showing the correlation coefficients between benchmarks and the corresponding LMArena categories.

For benchmarks within QA, Mathematics, and Code categories, the figures suggest a weak correlation between benchmarks' release date and their correlation with the corresponding LMArena category. For example, in the LMArena Overall subfigure, GPQA (released Nov 2023; $\rho = 0.79$) surpasses many subsequent benchmarks. In contrast, FACTS Grounding (released Jan 2025; $\rho = 0.32$) is inferior to those of earlier benchmarks. Conversely, in LMArena's Instruction Following subfigure, a positive trend between the correlation coefficients and the initial release dates is visible. Newer benchmarks like Creative Writing v3 (released Feb 2025; $\rho = 0.84$) show substantially higher correlation than older ones like IFEval (released Nov 2023; $\rho = 0.45$). These findings highlight a critical gap: While Alignment benchmarks are are being improved to better resemble human perception of LLM outputs, QA, Mathematics, and Code benchmarks show no such progress. We call for future benchmarks within these categories to be more aligned with human perception.

**Conclusion:** LLM performance on newer benchmarks is not necessarily more similar to human perception, except for benchmarks under the Alignment category. Alignment benchmarks are being improved to better resemble human perception of LLM outputs.

Table 4: Average Spearman correlation coefficients ($\rho$) between ranking on the full set of 24 benchmarks and ranking on each of the four categories in LMArena.

|  | LMArena's Overall Ranking | LMArena's Math Ranking | LMArena's Coding Ranking | LMArena's Instruction Following Ranking |
|---|---|---|---|---|
| Average | 0.69 | **0.78** | 0.69 | 0.69 |

## 4.4 Average Correlation Coefficients Across LMArena Categories

For each task-specific category in LMArena, Table 4 computes the average correlation coefficients across all associated benchmarks. This table is based on the schematic shown in Table 2, presenting the average of each column's $\rho$. We find that the average calculated using the Math category in LMArena ($\rho = 0.78$) is the highest. This indicates that existing benchmarks generally can better evaluate human perception when using LLMs for solving math problems.

**Conclusion:** Compared with other abilities (e.g., coding ability and instruction following ability), the math ability of LLMs can be relatively better evaluated by current benchmarks.

## 5 Discussion

In the previous analysis, we found that LLM performance on newer benchmarks within QA, Mathematics, and Code categories is not necessarily more similar to human perception. But we noticed $\tau^2$-Bench (Barres et al., 2025), which is a benchmark evaluating conversational agents in a dual-control environment, requiring coordination and communication between the agent and the user to succeed. This shows that capturing user perception has become an important target in the research community. But as this benchmark is the only one we found considering both reasoning ability and user perception, it's not suitable in any of our 4 categories.

Looking ahead, new benchmarks in categories such as Question Answering and Code that more directly capture user perception can be designed. For example, a benchmark in the Code category should focus more on the readability, conciseness, and their ability to make basic modifications to user-written code while providing instructional guidance. By explicitly incorporating user perception into task design and evaluation protocols, the benchmark will provide a more faithful measure of LLM utility in practice and help bridge the gap between benchmark scores and real-world user perception.

## 6 Conclusion & Limitations

We systematically compared LLM performance on popular benchmarks with human perception as reflected in LMArena. The results indicate that several benchmarks, such as FACTS Grounding and IFEval, do not resemble human perception. Moreover, neither benchmark difficulty nor recency reliably predicts a benchmark's resemblance to human perception. These findings highlight limitations in current benchmarking practices and the urgent need for more human-centered evaluation frameworks.

A key limitation of our study stems from LMArena itself. On LMArena, users often ask basic and simple questions, as more difficult or high-stakes problems are typically addressed directly using top-tier LLMs. Consequently, LMArena may not fully represent broader user perception. Additionally, some benchmarks include very few data points (e.g., IOI, with fewer than 15 LLMs being ranked on their leaderboards), which can introduce significant variability in correlation coefficients. For particularly difficult benchmarks where all LLMs achieve low accuracy, random factors may disproportionately affect performance, meaning small biases in accuracy can lead to large changes in LLM rankings and corresponding rank-order correlations. Finally, our categorization of benchmarks is necessarily approximate: most benchmarks evaluate multiple abilities. They cannot perfectly fit into a single broad category. For instance, Humanity's Last Exam primarily tests multi-subject reasoning, yet some of its problems are mathematical in nature. These limitations suggest that our results, while indicative, should be interpreted with caution.

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

## A APPENDIX

Table 5 lists Spearman correlation coefficients between each benchmark ranking and the ranking on each of the four task-specific categories in LMArena.

Table 6 lists the descriptions of our adopted benchmarks.

Table 7 and Table 8 list the URLs where we collect the scores and rankings on these benchmarks.

Table 9 lists the benchmarks we investigated but didn't adopt. We'll abandon a benchmark if it satisfies at least one of the descriptions below:

- The LLMs evaluated on the benchmark are too old that no LLMs ranking top 10 on the overall LMArena leaderboard are in its leaderboard.
- The benchmark has approached saturation, and no obvious distinction between benchmarks can be seen from the benchmark.

- The benchmark is too hard and there is also no obvious distinction.
- The benchmark evaluates complex ability, which cannot be categorized into our four categories.

Table 5: Spearman correlation coefficients ($\rho$) between each benchmark ranking and the ranking on each of the four task-specific categories in LMArena. This table expands the schematic shown in Table 2, presenting the full set of calculated Spearman correlation coefficients $\rho$. Rows group the 24 benchmarks by their primary category (QA, Mathematics, Coding, Alignment). Columns represent the four LMArena rankings (Overall, Coding, Math, Instruction Following). Each cell shows the $\rho$ value and its statistical significance (***$p < 0.001$, **$p < 0.01$, *$p < 0.05$; N.S. = Not significant). The bottom row shows the average correlation coefficient for each LMArena category across all benchmarks.

| | LMArena's Overall Ranking | LMArena's Math Ranking | LMArena's Coding Ranking | LMArena's Instruction Following Ranking |
|---|---|---|---|---|
| **QA** | | | | |
| SimpleQA | 0.79*** | 0.72*** | 0.73*** | 0.80*** |
| Facts Grounding | 0.32 N.S. | 0.45 N.S. | 0.23 N.S. | 0.27 N.S. |
| GPQA | 0.79*** | 0.93*** | 0.73*** | 0.74*** |
| MMLU-Pro | 0.86*** | 0.86*** | 0.87*** | 0.88*** |
| Humanity's Last Exam | 0.51 N.S. | 0.83*** | 0.43 N.S. | 0.35 N.S. |
| SuperGPQA | 0.92*** | 0.94*** | 0.92*** | 0.92*** |
| ARC-AGI-2 | 0.71** | 0.90*** | 0.66** | 0.71** |
| **Math** | | | | |
| MGSM | 0.64*** | 0.71*** | 0.65*** | 0.63*** |
| MATH-500 | 0.72*** | 0.89*** | 0.71*** | 0.69*** |
| FrontierMath (Tier 1-3) | 0.78*** | 0.87*** | 0.75*** | 0.78*** |
| AIME | 0.58*** | 0.78*** | 0.62*** | 0.48** |
| HMMT February 2025 | 0.67** | 0.85*** | 0.63** | 0.66** |
| **Coding** | | | | |
| HumanEval | 0.74*** | 0.84*** | 0.78*** | 0.79*** |
| LiveCodeBench | 0.78*** | 0.92*** | 0.77** | 0.66* |
| SWE-bench Verified | 0.68* | 0.54 N.S. | 0.80** | 0.72** |
| Aider polyglot | 0.80*** | 0.84*** | 0.87*** | 0.82*** |
| Terminal-Bench | 0.80*** | 0.77*** | 0.80*** | 0.82*** |
| SciCode | 0.68*** | 0.73*** | 0.68*** | 0.72*** |
| IOI | 0.76* | 0.83** | 0.64* | 0.78** |
| **Instruction Following** | | | | |
| IFEval | 0.34 N.S. | 0.39 N.S. | 0.31 N.S. | 0.45* |
| IFBench | 0.58*** | 0.70*** | 0.59*** | 0.61*** |
| ArenaHard | 0.73*** | 0.94*** | 0.80*** | 0.73*** |
| WritingBench | 0.71*** | 0.78*** | 0.79*** | 0.72*** |
| Creative Writing v3 | 0.80*** | 0.70*** | 0.79*** | 0.84*** |
| Average | 0.69 | 0.78 | 0.69 | 0.69 |

Table 6: Detailed Descriptions of Benchmark Datasets and Their Evaluation Focus

| Benchmark | Description |
|---|---|
| SimpleQA | Short, simple questions answering in broad domains. |
| FACTS Grounding | Responses generation based on provided documents, testing factuality especially in critical domains. |
| GPQA | Ultra-hard multiple-choice problems testing beyond-human knowledge. |
| MMLU-Pro | Reasoning, comprehension and memorizing abilities evaluation with 12K+ complex questions in 14 subjects. |
| Humanity's Last Exam | 2,500 Advanced reasoning test using 2500 extremely difficult questions over more than ten subjects. |
| SuperGPQA | Graduate-level questions reasoning covering niche professional fields. |
| ARC-AGI-2 | Evaluation of cognitive flexibility using grid-based puzzles, which requires ability of abstract reasoning, pattern cognition and generalization. |
| MGSM | Multilingual evaluation using translated GSM8K problems. |
| MATH-500 | 500-problem collection from MATH, spanning diverse mathematical knowledge. |
| FrontierMath | Challenging mathematical problems covering most major branches of modern mathematics. |
| AIME | Solving of high-difficulty competition problems covering algebra, geometry, number theory, and combinatorics. |
| HMMT February 2025 | Problems from an undergraduate-level math tournament. |
| HumanEval | A set of 164 hand-written programming problems for measuring functional correctness for synthesizing programs from docstrings. |
| LiveCodeBench | Code generation, repair, and comprehension on 300+ recent problems. |
| SWE-bench Verified | Benchmark of abilities to solve real-world software issues by evaluating agents' ability to generate a patch for given GitHub repository and issue description. |
| Aider Polyglot | Low-solve-rate coding problems from Exercism in six languages. |
| Terminal-Bench | Tasks for evaluating agents' performance in using terminals. |
| SciCode | Challenging problems from 16 diverse natural science fields to evaluate LMs' capabilities of code generation for the solution of real scientific research problems. |
| IOI | Problems from International Olympiad in Informatics |
| IFEval | Instruction-following capability testing with 500+ diverse prompts. |
| IFBench | Benchmark to evaluate precise instruction following generalization on 58 new, diverse, and challenging verifiable out-of-domain constraints. |
| ArenaHard | High-difficulty real-user queries testing ability of interacting. |
| WritingBench | Writing ability evaluation across 100+ article prompts. |
| Creative Writing v3 | Expansion of Creative Writing v2, using a hybrid rubric and Elo scoring system to enhance discrimination. |

Table 7: Details of the Sources of Accuracy Data on Benchmark Datasets

| Benchmark | Details |
| --- | --- |
| SimpleQA | Scores are obtained from https://www.kaggle.com/benchmarks/openai/simpleqa, and the last updated date of which is 2025/09/03. |
| FACTS Grounding | Scores are obtained from the column marked with caption "Score" in https://www.kaggle.com/benchmarks/google/facts-grounding, and the last updated date of which is 2025/08/20. |
| GPQA | Results are obtained from the "Overall" task type in https://www.vals.ai/benchmarks/gpqa-09-08-2025. |
| MMLU-Pro | Results are obtained from the "Overall" column in https://huggingface.co/spaces/TIGER-Lab/MMLU-Pro. |
| Humanity's Last Exam | Accuracy datas were fetched from https://agi.safe.ai/, last updated on 2025/04/03. |
| SuperGPQA | Results are obtained from the "Overall" column in https://supergpqa.github.io/. |
| ARC-AGI-2 | Results were obtained from column marked "ARC-AGI-2" in https://arcprize.org/leaderboard. |
| MGSM | Results are obtained from the "Overall" task type in https://www.vals.ai/benchmarks/mgsm-2025-09-08. |
| MATH-500 | Results are obtained from the "Overall" task type in https://www.vals.ai/benchmarks/math500-05-30-2025. |
| FrontierMath | Accuracy data are obtained from tab "Tier 1-3" in https://epoch.ai/frontiermath. Note that manual switching to the tab is required. |
| AIME | Results are obtained from the "Overall" task type in https://www.vals.ai/benchmarks/aime-2025-09-08. |
| HMMT Feburary 2025 | Accuracy data are obtained from tab "HMMT Feb 2025" in https://matharena.ai/?comp=hmmt–hmmt_feb_2025. |
| HumanEval | Results are obtained from the "HUMANEVAL" column in https://artificialanalysis.ai/leaderboards/models, note that the column will be visible only after clicking the "Expand Columns" button to the right of the page. |
| LiveCodeBench | Results are obtained from the "PASS@1" column in https://livecodebench.github.io/leaderboard.html, note that the time window should be manually adjusted to 5/1/2023-5/1/2025. |
| SWE-bench Verified | Data were obtained from tab "Bash Only" in https://www.swebench.com/. |
| Aider Polyglot | Results are obtained from the "Percent correct" column in https://aider.chat/docs/leaderboards/. |
| Terminal-Bench | Results are obtained from the "Overall" task type in https://www.vals.ai/benchmarks/terminal-bench-2025-09-18. |
| SciCode | Results are obtained from the "SCICODE" column in https://artificialanalysis.ai/leaderboards/models, note that the column will be visible only after clicking the "Expand Columns" button to the right of the page. |
| IOI | Results are obtained from the "Overall" task type in https://www.vals.ai/benchmarks/ioi-09-09-2025. |

Table 8: Details of the Sources of Accuracy Data on Benchmark Datasets (Continued)

| Benchmark | Details |
|---|---|
| IFEval | Results are obtained from the "IFEval - IFEval Strict Acc" column in https://crfm.stanford.edu/helm/capabilities/latest/#/leaderboard. |
| IFBench | Results are obtained from the "IFBENCH" column in https://artificialanalysis.ai/leaderboards/models, note that the column will be visible only after clicking the "Expand Columns" button to the right of the page. |
| ArenaHard | Results are obtained from the "Official Configuration" sector in README.md of https://github.com/lmarena/arena-hard-auto?tab=readme-ov-file#leaderboard. |
| WritingBench | Results are obtained from the "Overall" column in https://huggingface.co/spaces/WritingBench/WritingBench. |
| Creative Writing v3 | Results are obtained from column "Rubric Score" in https://eqbench.com/creative_writing.html. |

Table 9: Detailed Descriptions of Benchmark Datasets that were Examined but Not Used

| Benchmark | Description |
|---|---|
| MMLU | Multi-subject, broad-difficulty question answering, testing problem-solving depth and breadth. |
| BBH | Hard problems reasoning, enforcing CoT. |
| MuSR | Multistep soft reasoning tasks specified in a natural language narrative. |
| ZebraLogic | Non-monotonic logical reasoning under constraint satisfaction problems. |
| $\tau^2$-Bench | Evaluating conversational agents in a dual-control environment where both the agent and the user can operate tools, requiring coordination and communication between the agent and the user to succeed. |
| MATH | 12500 challenging competition math problems with step-by-step solution. |
| GSM8K | Elementary-level math word problems assessing basic reasoning. |
| MathBench | Progressive math from arithmetic to university-level, testing theoretical and applied skills. |
| MultiPL-E | Code generation over broad programming languages using problems in HumanEval and MBPP. |
| EvalPlus | Rigorous correctness testing using HumanEval and MBPP advanced by adversarial mutations. |
| CRUXEval | Code semantic understanding via output prediction and backward reasoning. |
| InfiBench | Code comprehension and correlated natural language QA in real context. |
| EvoEval | Evolution-augmented problems solving reducing overfitting. |
| MHPP | 210 Python problems with 7 core challenges testing NLP and edge-case handling. |
| BigCodeBench | Real-world challenging code completion and generation tasks. |
| AlpacaEval 2.0 | Advanced instruction-following benchmark with anti-gaming measures. |
| Creative Writing v2 | Text generation quality assessment through diverse prompts. |

