# OpenReview forum: "Benchmarking Large Language Model Benchmarks: Popular Benchmarks vs. Human Perception"
_ICLR.cc/2026/Conference — ICLR 2026 Conference Withdrawn Submission_

### Official Review · Reviewer_fSSB · 2025-10-21

**Soundness:** 2
**Presentation:** 2
**Contribution:** 3
**Rating:** 4
**Confidence:** 5

**Summary:**

This paper attempts to understand whether standard and popular benchmarks correlate with human perception, i.e., if the performance of these benchmarks is proportional to the "holistic goodness" and "overall" performance of an LLM. For this, the authors correlate the performance of an LLM on these benchmarks with the corresponding split from LMArena, which is an open leaderboard of LLMs as judged by humans via pairwise ranking (ELO score). The main comparisons are along QA, coding, math, and instruction-following. The main takeaway from this paper is that most of the popular and hard benchmarks reported with SOTA LLMs have low similarity with human perception.

**Strengths:**

- The paper addresses a very interesting problem of "whether standard benchmarks are enough for holistic evaluation of LLMs as perceieved  by humans". I personally appreciate this research direction, given the number of LLMs being produced with no solid and uniform holistic evaluation scheme that correlates with human perception and requirements.

**Weaknesses:**

- Overall, I find the paper quite poorly written, rather bland, and limited in both scope and analysis in its current form. It seems to contain a lot of filler content (e.g., lines 218–252), which provides redundant information more suitable for an appendix. I believe the authors could include more substantive core content if the paper were streamlined.

- Many of the hypotheses and claims in the paper rely heavily on the assumption of a "strong alignment" between LMArena and human perception across all tasks. While I acknowledge that the authors have mentioned this limitation and provided some justification, I believe that this assumption requires more corroboration.

- I am also surprised that the authors did not discuss benchmark contamination in Section 4.3 ([Ahuja et al., 2024a](https://aclanthology.org/2023.emnlp-main.258/); [Ahuja et al., 2024b](https://arxiv.org/pdf/2410.16186)). Contamination in LM-Arena is less likely due to its dynamic nature—new questions and tasks are continuously added—whereas standard benchmarks are static and more prone to such issues.

- Much of the paper’s results read like table-to-text descriptions of the correlation metric, without deeper analytical insights. The whole analysis and findings are based on the correlation between rankings from LMArena and standard benchmarks, which makes it unidimensional at the moment.

- I believe the paper would benefit from a more concrete analysis of contamination effects, inclusion of a broader range of models, comparisons across model sizes and families, and differentiation between “thinking” and “non-thinking” modes. Additional analysis supporting the conclusions of Section 4.1 would also be valuable, as prior studies have shown significant divergences between LLM-judge and human evaluations. In its current state, the paper feels more like a technical report than a comprehensive research study.

- Was there any analysis of overlap between LM-Arena tasks and standard benchmarks? If so, similar or intersecting problems should be excluded, as they could lead to inflated correlation scores.

- Finally, the paper lacks an analysis or visualization comparing the rankings of LLMs on standard benchmarks versus LM-Arena results (not with a simple correlation coefficient). Including such a comparison would strengthen the paper significantly.

**Questions:**

- It would be better to report the $R^2$ value for the regression lines in Figures 2 and 3 to provide a clearer measure of fit.

- The finding in Section 4.4 seems somewhat obvious, as math problems are objective and have a single fixed answer. Math problems in LMArena and any standard benchmark like GSM are very similar in terms of problem types. In this context, analyzing the intersection between LMArena and math benchmarks is crucial, as they may share some problems.

- Please provide the exact date on which LMArena was accessed to ensure reproducibility, given that the leaderboard and question bank are dynamic

- I am also unsure of the correlation comparison, as LMArena rankings are calculated via ELO (pairwise battles), while the paper just computes accuracies and ranks them.

---

> ### Author Response · Authors · 2025-11-26
>
> Thank you for your feedback. We plan to streamline the paper and elevate the analysis from simple correlation reporting to a structured investigation of why divergences occur.
> The interesting question is: Among the many Math benchmarks (HLE, AIME, MATH-500), which one correlates best with user experience in the "Math" category in LMArena, and why? Similarly, why might a particular Reasoning benchmark like GPQA diverge from user experience in the "Hard Prompts" category in LMArena?
>
> ### Proposed Methodology
>
> 1. Granular Category Alignment: We will perform strictly matched comparisons, such as comparing the LMArena Coding category against specific code benchmarks (e.g., LiveCodeBench), and comparing the LMArena Hard Prompts category against reasoning benchmarks.
> 2. Hypothesis Testing: We will analyze if factors like Prompt Strength (prompt length and complexity) and Output Type (generative vs. multiple choice) statistically predict the degree of alignment between a benchmark and user experience.
>
> ### Visualizations
>
> We will include the requested rank-comparison visualizations (scatter plots of Ranks vs. Ranks with regression lines) for these specific category pairs to make the alignment (or lack thereof) visually clear. An example is Figure 1 currently in the paper.

---

### Official Review · Reviewer_8oeM · 2025-10-31

**Soundness:** 2
**Presentation:** 2
**Contribution:** 1
**Rating:** 2
**Confidence:** 4

**Summary:**

The paper examines the correlation in model rankings between LMArena and 24 benchmarks. The authors compare the rank correlation of LMArena and (1) different benchmark categories (QA, math, code, and instruction following), (2) benchmark difficulty, and (3) benchmark release date.

**Strengths:**

The topic of the paper is interesting, the writing is reasonably clear, and, to the best of my knowledge this work is original in considering rank correlations for LMArena.

**Weaknesses:**

The main limitation of this work lies in the significance of its contributions and the soundness of its claims.
The authors show that the rankings of some popular benchmarks differ from those of LMArena. However, this is to be expected. As noted in line 127, the authors correctly point out that even benchmarks that appear very similar can exhibit substantial disagreement. If the authors had found that **most** popular benchmarks were uncorrelated with LMArena rankings, that would indeed be surprising and noteworthy. But the fact that **some** benchmarks differ is neither unexpected nor particularly significant. In fact, the strength of the correlations reported in Table 3 is, if anything, somewhat surprising.

Regarding Figure 2 (which I believe corresponds the missing Section 4.2), I think the authors have not identified the correct source of the low rank correlations. I hypothesize that HLE is uncorrelated not because it is difficult, but because all models are tightly clustered together (i.e., lack of resolution). The fact that this cluster happens to be centered around zero (due to the task’s difficulty) is less relevant than the clustering itself. The authors might obtain more insight by examining the mean performance difference between consecutive ranks as an alternative metric to regress.

Regarding Figure 3, I believe the findings are not robust, and the conclusions could change simply by excluding the very earliest benchmark. The trend for LMArena Code appears constant, but might appear increasing if HumanEval were removed. Conversely, the trend for IF appears increasing, but could appear constant if IFEval were excluded. The fact that removing a single point would alter the conclusion makes me worry of the soundness of the findings.

While the topic of the paper is very interesting, and I encourage the authors to pursue this line of work further, I believe that, at this stage, the significance and scope of the contributions do not warrant acceptance.

**Questions:**

Note: I disagree with the framing that LMArena reflects “human perception”. LMArena is also a benchmark, albeit one that arguably more closely resembles chat assistant-like use of models compared to most other benchmarks out there. But it certainly does not evaluate LLMs “in the wild” or human perception in broad sense, it is very much a controlled environment where users go for the explicit purpose of evaluating LLMs.

It would be interesting to compare the average correlation between LMArena and the different benchmark categories, to the average correlation between the benchmarks in each of the categories (e.g., mean correlation between LMArena math and Benchmarks math against mean correlation between Benchmarks math).

Much of the content in pages 3, 4, and 5 would be better suited for the Appendix.

Typos:
MMLU in L32

---

> ### Author Response · Authors · 2025-11-26
>
> Thank you for your insightful review. We agree that finding "some benchmarks don't correlate" is, by itself, an expected and essentially negative result. We also appreciate your hypothesis that "clustering/lack of resolution" (rather than difficulty per se) might be the cause. We have refined our research plan to directly address these points and produce a positive, explanatory result.
>
> We want to move from "Checking if they correlate" to "Explaining what makes a benchmark predictive of user experience." Among benchmarks testing similar capabilities, why do some diverge from human usage patterns (LMArena) while others do not?
>
> We propose to test a set of specific hypotheses to identify the characteristics of a "good" benchmark (defined as one that predicts user experience).
> - The Variance Hypothesis: You hypothesized that HLE is uncorrelated due to clustering (low resolution). We will explicitly calculate the variance of model scores within each benchmark and test if score variance (standard deviation of model scores within the benchmark) is a significant predictor of Spearman correlation with LMArena.
> - The Difficulty Hypothesis: We will re-evaluate the difficulty hypothesis by controlling for variance.
> - The Format Hypothesis: We hypothesize that benchmarks requiring generative answers align better with LMArena than multiple-choice tasks.
>
> We will perform these analyses by mapping benchmarks strictly to their LMArena counterparts (e.g., comparing Humanity’s Last Exam / AIME rankings only with LMArena’s Math ranking of LLMs).

---

### Official Review · Reviewer_6WXW · 2025-11-03

**Soundness:** 2
**Presentation:** 2
**Contribution:** 1
**Rating:** 2
**Confidence:** 4

**Summary:**

This paper attempts to "benchmark the benchmarks" by comparing LLM rankings on 24 popular academic benchmarks against the crowd-sourced rankings from the LMArena platform. The authors use LMArena as a 'ground truth' for 'human perception'. They find that many specialized benchmarks (e.g., FACTS Grounding, Humanity's Last Exam) show low or insignificant correlation with LMArena rankings. They conclude that many benchmarks are misaligned with human perception and that neither difficulty nor recency usually guarantees alignment.

**Strengths:**

The paper's motivation is valid. It targets an increasingly important issue, which is to investigate whether benchmark performance translates into user-perceived quality, as LLM research shifts from leaderboard optimization to practical usability. The inclusion of 24 benchmarks and 114 LLMs is also impressive.

**Weaknesses:**

1. The paper considers that LMArena represents Human Perception. Which is a flawed assumption. LMArena assesses how humans comparatively favor models when evaluated head-to-head in dialogue-based interactions. Therefore, it is completely unsurprising that a model's ability to pass a graduate-level exam (SuperGPQA) or solve complex reasoning problems (Humanity's Last Exam) does not correlate with its ability to be a more engaging or helpful chatbot for simple queries.

2. Moreover, the comparison methodology is fundamentally broken.

i. FACTS Grounding (tests factual grounding in long documents) is compared to LMArena's overall ranking, which is dominated by general chat quality. Why would these correlate?

ii. Humanity's Last Exam (extremely difficult reasoning) is compared to LMArena, where users ask simple questions, which the authors also discuss in the Limitations section.

3. No confidence intervals reported on correlations.

4. The work seems quite similar to what Laskar et al. (https://aclanthology.org/2024.emnlp-main.764/) have done in Table 3. However, no citations have been provided. This also makes the novelty very limited.

**Questions:**

1. Justify how LLMArena represents human perception?
2. Justify how the comparison methodology is correct (e.g., how FACTS grounding or Humanity's last exam benchmarks could correlate with LMArena)
3. What correlation coefficient would you consider good enough? and Why?

---

> ### Author Response · Authors · 2025-11-26
>
> ### Response to questions 1 and 2
>
> Thank you for your comprehensive review. We understand your concern that comparing specialized benchmarks (like FACTS or HLE) to LMArena’s "Overall" ranking is a mismatched comparison (or "apples to oranges"). We agree that this was a flaw in our initial framing and have used this valid critique as the foundation for a proposed new direction.
>
> We seek to determine: **Among benchmarks testing similar capabilities, which ones reflect performance that aligns with the actual user experience in that specific domain?**
>
> For example, while it is unfair to compare HLE (reasoning) with LMArena Overall (chat), it is scientifically meaningful to compare HLE and AIME rankings specifically against LMArena’s Math or Hard Prompts rankings.
>
> We propose to refine our methodology by introducing granular comparisons:
> - **Hard vs. Hard:** Comparing Humanity’s Last Exam (HLE) and GPQA specifically against the **LMArena "Hard Prompts" Category** (which filters for complex, problem-solving queries).
> - **Math vs. Math:** Comparing AIME and FrontierMath specifically against the **LMArena Math Category**.
>
> We aim to rigorously demonstrate whether the previously observed misalignment persists when we align the task categories and difficulty levels as closely as possible.
>
> In addition, we aim to go beyond merely reporting "weak correlations" by identifying the factors that predict high correlation. We will test several hypotheses, such as:
> - **Difficulty:** Even within the "Hard Prompts" category, does extreme benchmark difficulty inversely correlate with user preference?
>
> ### Response to question 3
>
> We appreciate your insightful question: "What correlation coefficient would you consider good enough, and why?" We would like to clarify that our study focuses on the relative magnitude of correlation coefficients and the factors influencing them—such as benchmark difficulty and recency—rather than establishing a specific absolute threshold for sufficiency.

---

### Official Review · Reviewer_xVZQ · 2025-11-04

**Soundness:** 2
**Presentation:** 2
**Contribution:** 1
**Rating:** 2
**Confidence:** 4

**Summary:**

The paper examines how correlated popular LLM benchmarks agree with human perception by computing the Spearman correlation between the leaderboard LLM rankings from 24 benchmarks and task-matched rankings on LMArena. Several QA/alignment benchmarks have low or non-significant rank correlation with LMArena. Also, benchmark difficulty and recency mostly don't predict higher correlation.

**Strengths:**

- A clear, focused research question of interest using a simple, reproducible method (Spearman's correlation on overlapping model sets)
- It broadly used 24 benchmarks across QA/Math/Code/Alignment areas and 4 LMArena categories

**Weaknesses:**

- The paper acknowledges several limitations of using only LMArena as a standalone human perception dataset. It is plausible but still incomplete. LMArena tends to consist of casual, English queries. In the LMArea leadeboards, top models receive disproportionate exposure, and prompt mix varies over time. The paper notes this, but it still interprets correlations as "resemblance of human perception" rather than 'LMArena-specific preferences." Some level of claim tempering is needed.

- The paper reports 96 correlation scores (24 benchmarks x 4 LMArena categories) with significance, but it doesn't control for multiple comparisons or show any uncertainty on the correlation scores (e.g., no 95% CIs, no bootstrap variability, no sensitivity to model-set perturbations). The paper currently only provides statistics for significance and one p-value per pair.

-  Only one rank-agreement metric. Alongside Spearman, report Kendall's tau, top-k overlap (k=5 or 10), and RBO to conduct a robustness check on your correlation results. These are important as some benchmarks have small Ns. I would recommend using RBO as it doesn't require two ordered lists of the same length.

- The paper assembles benchmark scores and rankings from heterogeneous leaderboards and then computes correlation between rankings after intersecting model sets (common models in both ordered lists). Differences in scoring rules, prompt formats (if LLM-as-judge), and date of entries, etc can induce spurious rank differences. The authors did not provide details of how they employ fair rules in such cases.

**Questions:**

- Please address the above weaknesses.
- Include other human preference datasets or curated ones in order to make the paper claim generalizable.

---

> ### Author Response · Authors · 2025-11-26
>
> Thank you for your constructive feedback. We appreciate your point regarding the limitations of LMArena as a proxy for "human perception" and the need for more robust statistical rigor.
>
> Instead of treating LMArena as a monolithic ground truth, we plan to treat it as a representation of **"the actual user experience in specific domains"** (as proxied by corresponding task-specific user-prompt categories in LMArena). For instance, we plan to determine: Does the performance ranking on HLE (Humanity's Last Exam) align with user experience of model capability when asking university-level/hard questions in LMArena? Currently, multiple benchmarks often assess similar capabilities (e.g., GPQA and HLE both evaluate model performance on extremely difficult questions). Our refined research interest is to identify, among benchmarks designed to test similar capabilities, which ones reflect performance that aligns with the user experience in their specific domains, and which ones diverge.
>
> Specifically, we propose to restructure the study around a set of explanatory hypotheses to analyze the factors driving the gap between benchmarks and LMArena rankings.
>
> When testing each hypothesis, we will systematically compute Spearman's rank correlation, Kendall's tau, and RBO (Rank-Biased Overlap) to cross-verify our findings, thereby ensuring that our conclusions are robust and not dependent on the choice of a particular correlation metric, such as Spearman's rank correlation.
>
> Our proposed analysis would test:
> * **The Variance Hypothesis:** Benchmarks with higher score variance across models exhibit higher correlation with User Experience.
> * **The Difficulty Hypothesis:** When variance is controlled, benchmarks with higher difficulty (lower average model scores) generally exhibit lower correlation with User Experience, even when benchmarks are compared with the "Hard Prompts" category in LMArena.
> * **The Format Hypothesis:** Benchmarks requiring open-ended text generation exhibit higher correlation with User Experience than multiple-choice benchmarks.
> * **The Prompt Complexity Hypothesis:** Benchmarks with longer and more complex prompt structures exhibit higher correlation with User Experience.
> * **The Scale Hypothesis:** Benchmarks with a larger number of questions exhibit higher correlation with User Experience.
>
> We would also be grateful for any additional hypotheses you might suggest to further enrich this exploratory analysis.

---

> > ### Comment · Reviewer_xVZQ · 2025-11-26
> >
> > Thank you for the clarifications that you've made within this short timeframe. However, I will maintain my original score, as I believe these explanations are critical and should have been included in the submitted manuscript.

---

### Author Response · Authors · 2025-11-26
**Pivot to "User Experience" and Hypothesis-Driven Factor Analysis**

We thank all reviewers for their constructive and detailed feedback. We recognize the consensus that simply reporting low correlations is insufficient and that our terminology regarding "human perception" was imprecise.

Based on your valuable suggestions, we plan to restructure our research methodology. We are shifting the focus from a descriptive correlation report to a **hypothesis-driven investigation** into *why* certain benchmarks align with human preference while others do not.

Below is our revision plan:

## 1. Re-scoping from "Human Perception" to "User Experience"
We agree with Reviewers **6WXW** and **8oeM** that LMArena does not represent all of "human perception" but rather a specific mode of interaction. We will replace the term "Human Perception" with **"User Experience (UX)."**

We define **User Experience** in this context as "the baseline satisfaction of human users interacting with LLMs within a specific domain (e.g., Math, Coding)." This definition aligns with the LMArena categorization (as detailed in the LMArena blog), which reflects how users objectively rate models when performing distinct tasks (e.g., coding, creative writing).

## 2. Methodology Update
To address the fairness concerns raised by Reviewer **6WXW** (e.g., comparing FACTS Grounding to LMArena Overall ranking of LLMs), we will strictly align benchmarks to their corresponding specific LMArena categories. We will perform two types of comparisons:
1.  **Domain-specific comparison:** Strictly comparing Benchmark $X_{domain}$ with LMArena Category $Y_{domain}$.
2.  **Global baseline comparison:** Comparing all benchmarks against LMArena Overall to test generalizability.

If a benchmark does not fit any category, it will be excluded from Domain-specific comparison.

| LMArena Category | Definition | Aligned Benchmarks |
| :--- | :--- | :--- |
| **Hard Prompts** | Complex reasoning, domain knowledge, problem-solving. | **Humanity's Last Exam (HLE)**, **GPQA**, **MMLU-Pro**, **FrontierMath** |
| **Math** | Calculation, algebra, logical deduction. | **AIME**, **MATH-500**, **HMMT Feb 2025**, **MGSM** |
| **Coding** | Generation, debugging, explanation. | **LiveCodeBench**, **HumanEval**, **Aider Polyglot**, **StructEval**, **EffiBench-X** |
| **Instruction Following** | Constraints, formatting, length control. | **IFEval**, **IFBench**, **AlpacaEval 2.0** |

## 3. Hypothesis Testing
We agree with Reviewer **8oeM** that finding *some* benchmarks do not correlate is, by itself, unsurprising. However, identifying the **determinants** of alignment is scientifically valuable. We will move beyond simple correlation reporting to a hypothesis testing framework. We will use statistical models to test the following declarative hypotheses (H1-H6) to identify the features that make a benchmark predictive of User Experience.
* **H1 (The Difficulty Hypothesis):** Benchmarks with higher difficulty (lower average model scores) generally exhibit lower correlation with User Experience, even when benchmarks are compared with the "Hard Prompts" category in LMArena.
* **H2 (The Recency Hypothesis):** Benchmarks released more recently exhibit higher correlation with User Experience. This tests the impact of dataset contamination and the evolution of benchmarks to match current model capabilities.
* **H3 (The Prompt Complexity Hypothesis):** Benchmarks with longer and more complex prompt structures exhibit higher correlation with User Experience.
* **H4 (The Generative Hypothesis):** Benchmarks requiring open-ended text generation exhibit higher correlation with User Experience than multiple-choice benchmarks.
* **H5 (The Variance Hypothesis):** Benchmarks with higher score variance across models exhibit higher correlation with User Experience.
* **H6 (The Scale Hypothesis):** Benchmarks with a larger number of test items exhibit higher correlation with User Experience.

## 4. Statistical Rigor
We will expand our correlation analysis beyond Spearman’s $\rho$. We will include **Kendall’s $\tau$** (for ordinal association) and **Rank-Biased Overlap (RBO)** (to weigh the top of the leaderboard more heavily, addressing the "top-tier" usage focus). To test the hypotheses, We will use mixed-effects regression models to verify which of the factors (H1-H6 variables) are statistically significant predictors of the correlation coefficient.

We believe this hypothesis-driven approach can transform the study from a report into an analysis of what constitutes a user-aligned benchmark. We welcome any further thoughts on these specific hypotheses.

Best regards,

The Authors

---

### Note · Authors · 2026-01-08

I have read and agree with the venue's withdrawal policy on behalf of myself and my co-authors.